# Inhibiting Intracellular α_2C_-Adrenoceptor Surface Translocation Using Decoy Peptides: Identification of an Essential Role of the C-Terminus in Receptor Trafficking

**DOI:** 10.3390/ijms242417558

**Published:** 2023-12-16

**Authors:** Aisha Raza, Saima Mohsin, Fasiha Saeed, Syed Abid Ali, Maqsood A. Chotani

**Affiliations:** 1Dr. Panjwani Center for Molecular Medicine and Drug Research (PCMD), International Center for Chemical and Biological Sciences, University of Karachi, Karachi 75270, Pakistan; aisha.raza993@gmail.com (A.R.); saimamohsin449@gmail.com (S.M.); fasihasaeed37@gmail.com (F.S.); 2Husein Ebrahim Jamal (H.E.J.) Research Institute of Chemistry, International Center for Chemical and Biological Sciences, University of Karachi, Karachi 75270, Pakistan; dr.syedabidali@gmail.com

**Keywords:** microvascular, α_2C_-adrenoceptors, vasoconstriction, protein–protein interactions, Rap1A GTPase, receptor translocation, decoy peptides

## Abstract

The G protein-coupled α_2_-adrenoceptor subtype C (abbreviated α_2C_-AR) has been implicated in peripheral vascular conditions and diseases such as cold feet–hands, Raynaud’s phenomenon, and scleroderma, contributing to morbidity and mortality. Microvascular α_2C_-adrenoceptors are expressed in specialized smooth muscle cells and mediate constriction under physiological conditions and the occlusion of blood supply involving vasospastic episodes and tissue damage under pathological conditions. A crucial step for receptor biological activity is the cell surface trafficking of intracellular receptors, triggered by cAMP-Epac-Rap1A GTPase signaling, which involves protein–protein association with the actin-binding protein filamin-2, mediated by critical amino acid residues in the last 14 amino acids of the receptor carboxyl (C)-terminus. This study assessed the role of the C-terminus in Rap1A GTPase coupled receptor trafficking by domain-swapping studies using recombinant tagged receptors in transient co-transfections and compared with wild-type receptors using immunofluorescence microscopy. We further tested the biological relevance of the α_2C_-AR C-terminus, when introduced as competitor peptides, to selectively inhibit intracellular α_2C_-AR surface translocation in transfected as well as in microvascular smooth muscle cells expressing endogenous receptors. These studies contribute to establishing proof of principle to target intracellular α_2C_-adrenoceptors to reduce biological activity, which in clinical conditions can be a target for therapy.

## 1. Introduction

Raynaud’s syndrome (RS) is a vasospastic disorder of the microvasculature triggered by cold or emotional stress [1,2]. RS is categorized as idiopathic or primary Raynaud’s syndrome (PRS) when no underlying reason for this impairment is identified, and secondary Raynaud’s syndrome (SRS) when it is associated with a pre-existing or multiple diseased condition [3]. About 5% of the general population is affected by RS [4], and this number increases in areas where the climate is comparatively colder [5,6,7]. RS has a very high prevalence rate in patients with scleroderma and systemic sclerosis [3]. The onset of RS physically manifests with a triphasic coloration of the digits [8,9], followed by swelling, numbness, and lesions [10].

RS is known to be a multifactorial disease and can be caused by defective endothelial vasodilation caused by reduced levels of calcitonin gene-related peptide (CGRP) [11] resulting in a drop in the production of vasodilator nitric oxide [12]. One of the sole causes of RS seems to be a default in the vascular thermoregulatory system [13,14], especially in the digits where lower temperatures can significantly reduce blood flow in not only in the arteriovenous anastomosis but also in the nutritional flow in the capillaries due to the increased sympathetic activation of α_2_-adrenoceptors in vascular smooth muscle cells [15]. 

The α_2_-adrenoceptors belong to the family of G protein-coupled receptors (GPCRs), and pharmacologically, three subtypes of α_2_-adrenoceptors have been identified; α_2A_, α_2B_, and α_2C_ [16]. Initially, studies conducted in the intact cardiovascular system of mice showed that for α_2_-adrenoceptors, only subtype α_2A_ and α_2B_ possessed physiological roles [17,18], whereas α_2C_ was found unresponsive or silent [19]. However, Chotani and colleagues further investigated and disclosed that at 37 °C, the α_2C_-adrenoceptor remained non-functional as found in earlier studies, but at 28 °C, the α_2C_-adrenoceptor became activated and could generate a cold-induced vasospastic episode in RS [20].

Studies have found that α_2C_-adrenoceptors are primarily localized in the ER or Golgi compartment of the cell [21,22,23] at 37 °C and translocate to the cell surface when the temperature is lowered to 28 °C [20,24]. The α_2C_-adrenoceptors have been found to contain an arginine-rich region (RRRRR) in the carboxyl (C)-terminus, known to be a retention signal for the ER [25], which may explain its perinuclear localization in the cell. Several signaling pathways mediate the translocation of α_2C_-adrenoceptors toward the cell surface. Upon the selective pharmacological inhibition of Rho kinase by fasudil at 28 °C and of COX-2 by celecoxib, a decrease in the expression and translocation of these receptors has been observed [23,26,27]. cAMP has a dual effect on the regulation of α_2C_-adrenoceptors, augmenting its transcription through the JNK-AP1 pathway [28] and redistributing it to the cell surface through RhoA-Rho-kinase using the cytoskeletal actin-binding protein filamin-2 through Rap1A GTPase intracellular signaling [29]. Filamin-2 possesses a conserved serine whose phosphorylation has been reported to be a key player in the reconstruction of the cytoskeleton [30,31]. The amino acids in the C-terminus of α_2C_-adrenoceptors interact with a 78 amino acid sequence motif of filamin-2 containing the conserved serine, to translocate to the cell surface. 

In this study, we have assessed the role of the α_2C_-adrenoceptor C-terminus in receptor cell surface translocation and have utilized the α_2_-adrenoceptor subtype A (α_2A_-adrenoceptor), which does not associate with filamin-2, for comparison. These studies provide a unique insight into the role of the C-termini of both α_2_-adrenoceptor subtypes in regulating receptor cellular localization and allow to experimentally test the application of using the C-terminal domain to target receptor trafficking. 

## 2. Results

### 2.1. Transient Co-Transfections

The α_2C_-adrenoceptors have a unique mechanism of regulation; intracellular receptors translocate to the cell surface in response to extracellular stimuli, including signaling which increases intracellular levels of cAMP, which was established in our previous studies utilizing primary cultures of microvascular smooth muscle cells (microVSM) derived either from murine tail artery or human dermal arterioles [29]. For example, see Figure 1 for the effect of the adenylyl cyclase activator forskolin (which activates intracellular cAMP-Epac-Rap1A signaling) on endogenous α_2C_-adrenoceptor translocation when added to quiescent primary murine microVSM, similarly seen in human-derived microVSM [29]. These cell-surface-associated α_2C_-adrenoceptors were shown to be functional in both murine- and human-derived microVSM [29,32].

We initially tested recombinant DNA expressing α_2C_-adrenoceptor fused to green fluorescent protein (α_2C_-AR-GFP) in primary cultures of murine microVSM. For example, see Figure 2 for the localization of α_2C_-AR-GFP in quiescent, unstimulated microVSM, which is similar to endogenous α_2C_-adrenoceptors under quiescent conditions (compared with localization in DMSO-solvent control cells, Figure 1), which illustrates the feasibility of using GFP-fused or other HA-tagged constructs for transient transfection studies. However, due to the extremely low transfection efficiency in these primary cultures (<1%, assessed by determining the number of GFP-positive cells), further studies on recombinant DNA constructs could not be conducted in microVSM. Alternatively, we utilized and optimized a mouse-embryo-derived fibroblast (NIH/3T3) cell line for transient transfections/co-transfections. NIH/3T3 cells were relatively fast growing and showed better transfection efficiency and were therefore utilized for further studies.

Transient co-transfections were performed in NIH/3T3 cells utilizing recombinant DNA tagged constructs along with a constitutively active version of the small GTPase Rap1A (Rap1A-63E), compared with the control backbone expression vector pcDNA. Indeed, the results showed the recapitulation of intracellular signaling associated with α_2C_-adrenoceptors in NIH/3T3 cells compared with the Rap1A-α_2C_-adrenoceptor coupled signaling and cell surface receptor trafficking in human and murine microVSM [29], for receptors tagged either with HA (Figure 3) or with GFP (Figure 4). In these transfections, in comparison, we noted differences in signal fluorescence intensities between cells labeled with primary and secondary antibodies (higher fluorescence) versus the direct detection of fluorescence in cells expressing green fluorescent protein (relatively lower fluorescence). Further studies were therefore carried out in NIH/3T3 cells to test the role of the C-termini in receptor trafficking coupled to Rap1A GTPase intracellular signaling, utilizing the receptor chimeras described in Section 4.

### 2.2. Domain-Swapping Studies

To test the biological relevance of C-termini in Rap1A GTPase coupled trafficking, receptor chimeras were generated for comparison with wild-type receptors. These are described in detail in Section 4.

#### 2.2.1. α_2A_-Adrenoceptor Chimera Studies

Transient co-transfection studies showed the wild-type α_2A_-adrenoceptors to be present primarily on the cell surface in cells with the control expression vector pcDNA. However, the localization of α_2A_-adrenoceptors changed significantly (*p* < 0.0001) in response to Rap1A GTPase signaling, with a reduction in cell surface receptors (Figure 5A,B,E).

This profile changed dramatically when the α_2C_-adrenoceptor C-terminus was swapped with the wild-type α_2A_-adrenoceptor C-terminus region. The chimera α_2A_-_2C_-AR was primarily intracellular, similar to α_2C_-adrenoceptor wild-type receptors, when co-transfected with the control expression vector pcDNA. However, in cells co-transfected with activated Rap1A, the α_2A_-_2C_-AR chimera showed a significant association with the cell surface (*p* < 0.0001), similar to wild-type α_2C_-adrenoceptors (Figure 5C–E).

#### 2.2.2. α_2C_-Adrenoceptor Chimera Studies

Transient co-transfection studies showed the wild-type α_2C_-adrenoceptor to be primarily intracellular with the control expression vector pcDNA. However, the localization of α_2C_-adrenoceptors changed significantly (*p* < 0.0001) in response to Rap1A GTPase signaling, with an increase in cell surface receptors (Figure 6A,B,E).

This profile of wild-type α_2C_-adrenoceptors changed dramatically when the α_2A_-adrenoceptor C-terminus was swapped with the wild type α_2C_-adrenoceptor C-terminus region. The chimera α_2C_-_2A_-AR was primarily on the cell surface, similar to α_2A_-adrenoceptor wild-type receptors, when co-transfected with pcDNA. However, in cells co-transfected with activated Rap1A, the α_2C_-_2A_-AR chimera was significantly reduced on the cell surface (*p* < 0.0001), similar to wild-type α_2A_-adrenoceptors (Figure 6C,D,E).

### 2.3. Peptide Expression, Purification, and Detection by Western Blotting

The peptides were expressed in a BL21 (DE3)-STAR bacteria strain and purified as described in Section 4. 

The peptides contain the HA-tag, which was used for the confirmation of expression and the purification in the Western blot analysis (Figure 7).

#### 2.3.1. Peptide Delivery and Localization in NIH/3T3 Cells

Intracellular delivery and the specificity of peptides (encoding α_2C_-AR^449–462^ target sequence and control sequence for α_2_-adrenoceptor subtype A, α_2A_-AR^438–450^, 25 µL each of purified peptide) were tested in transient co-transfection studies utilizing α_2C_-AR-GFP/Rap1A-CA and anti-HA antibody for peptide detection using immunofluorescence microscopy.

The effect on GFP-positive cells with peptide uptake was assessed. Our observations showed that the peptide staining was distinct from the background signal, which was assessed for cells without peptides and with or without primary anti-HA antibody (Figure 8A–D). Using this approach, we detected the cellular uptake of both peptides, which showed localization in the cytosol or were perinuclear, Figure 8E–G for pTAT–HA–A2A–AR^438–450^ (Peptide A) and Figure 8H–J for pTAT–HA–A2C–AR^449–462^ (Peptide C).

#### 2.3.2. Peptide Specificity to Inhibit Receptor Translocation in NIH/3T3 Cells

Based on the outcomes of the domain-swapping studies, we hypothesized that the peptide encoding α_2C_-AR^449–462^ should selectively inhibit α_2C_-adrenoceptor trafficking. 

Indeed, upon quantification of cell-surface-associated α_2C_-AR-GFP, Rap1A stimulation significantly increased receptor translocation compared with control expression vector pcDNA (*p* < 0.0001), which was not significantly affected in the presence of intracellular Peptide A (*p* = *ns*). However, in the presence of intracellular Peptide C, the α_2C_-AR-GFP receptor translocation was selectively and significantly (*p* < 0.0001) reduced (Figure 9).

#### 2.3.3. Peptide C Effect on Endogenous microVSM α_2C_-Adrenoceptor Translocation and Function 

Based on the outcomes of transient transfection studies, Peptide C was further tested in human-arteriolar-derived microVSM which express α_2C_-adrenoceptors [27]. In these cells, adenylyl cyclase activation by forskolin activates cAMP-Epac-Rap1A signaling which is coupled to the increased expression and translocation of α_2C_-adrenoceptors through the RhoA-Rho kinase pathway and receptor trafficking via an association with the actin-binding protein filamin-2 [29,33]. Alternatively, the Epac-Rap1A signaling can also be activated by the cAMP analog 8-pCPT-2′-O-Me-cAMP [29,33]. In these microVSM, the cAMP analog 8-pCPT-2′-O-Me-cAMP increased surface α_2C_-adrenoceptors compared with the solvent control cells (Figure 10A,E, *p* < 0.0001). However, in the presence of Peptide C, the cell surface localization of the receptor was significantly reduced (*p* < 0.0001) and was below the baseline level in unstimulated cells (Figure 10B–E, *p* < 0.0001). The receptor and Peptide C co-localized in the perinuclear region of microVSM (Figure 10C,D). These observations were further validated by assessing microVSM receptor function by measuring the intracellular levels of cAMP in 8-pCPT-2′-O-Me-cAMP (100 µM, 14 h) stimulated cells in the absence or presence of Peptide C. Since the α_2_-adrenoceptors are Gi-coupled, activation by the agonist UK, 14,304 inhibits adenylyl cyclase and intracellular cAMP levels [24,29,33]. Indeed, these studies supported the loss of cell surface receptor functional activity (Figure 10F). In the absence of this inhibition, intracellular levels of cAMP increased noticeably.

## 3. Discussion

There has been tremendous interest in understanding the physiological mechanisms involved in the extreme temperature sensing and avoidance ability in humans pertaining to survival and protection from tissue damage. This includes the recent identification of cold sensor receptors in the sensory nerves of the skin, which has added to a better understanding of the mechanisms involved in thermoregulation [34]. Particularly, interest in inhibiting the biological activity of vascular α_2C_-adrenoceptors (which play a role as thermo-effectors) has grown over the years and includes the recent elucidation of crystal structure to identify molecular interactions during ligand binding which can aid in designing selective α_2C_-adrenoceptor antagonists [35]. Indeed, clinical trials have been performed using α_2C_-adrenoceptor antagonists targeting cell surface receptors and include compounds OPC-28326, ORM-12471, and BAY1193397 [36,37]. However, since this receptor is also expressed on the cell surface in non-VSM cells, other tissues, and organs, this approach, although promising in short-term trials, also showed an effect on inhibiting pre-synaptic α_2C_-adrenoceptors and in augmenting sympathetic activity over long-term use, which affects heart function [38]. There are therefore impediments to clinical progress as there is no targeted therapy for vasospasms involving α_2C_-adrenoceptors, for example, in Raynaud’s phenomenon.

Consequently, therapies are focused on improving blood circulation and include the use of vasodilators, including calcium channel blockers, angiotensin-converting enzyme (ACE) inhibitors to reduce the number, duration, and severity of attacks and to reduce the inflammation using cyclooxygenase-2 (COX-2) inhibitors [39,40,41,42]. For severe Raynaud’s non-responsive to medication, digital sympathectomy, which involves cutting the sympathetic nerves going to the blood vessel in the fingers, is a surgical option [7]. Therefore, new approaches selectively targeting α_2C_-adrenoceptor translocation and surface expression in early stages of the disease are necessary for better clinical outcomes.

The work performed by Chotani and colleagues utilizing human arteriolar smooth muscle cells identified a novel molecular mechanism of α_2C_-adrenoceptor cell surface translocation at a physiological temperature of 37 °C, which is coupled to the intracellular second messenger cyclic AMP (cAMP). cAMP, through the guanine exchange factor Epac, activates the Ras superfamily small GTP-binding protein Rap1A which mobilizes functional α_2C_-adrenoceptors through Rho-Rho kinase ROCK and filamentous (F) actin to the cell surface [29]. Using experimental yeast two-hybrid genetic screen and in silico approaches, it was determined that the carboxyl terminus’ last 14 amino acids (amino acids 449–462 in the full length protein) are crucial for protein–protein interactions with the actin cross-linker protein filamin-2 amino acids [33,43]. Critical residues were identified in the α_2C_-adrenoceptor involved in filamin-2 interaction, which include amino acids 449, 454, 456, and 461 [43]. Research has suggested that targeting the specific interaction of α_2C_-adrenoceptors with the actin cytoskeleton can be of high therapeutic importance in order to treat people suffering from Raynaud’s syndrome [44]. 

In the current study, we have built on these findings and have further validated the role of the carboxyl terminus using independent approaches, including receptor chimeras and competitor/decoy peptides. The outcomes of these studies show that the carboxyl termini of α_2_-adrenoceptor subtypes A and C play a major role in receptor localization and trafficking. Specifically, the subtypes showed different localization under basal, unstimulated conditions, with the α_2A_-adrenoceptor subtype predominantly on the cell surface, and the α_2C_-adrenoceptor subtype predominantly intracellular. However, upon Rap1A stimulation, cell surface α_2A_-adrenoceptors were reduced, whereas α_2C_-adrenoceptors were increased. Rap1A signaling, therefore, appears to have opposing effects on α_2_-adrenoceptor subtype localization and cell trafficking, increasing the cell surface α_2C_-adrenoceptor subtype, and at the same time, reducing the cell surface α_2A_-adrenoceptor subtype. 

In our previous studies, we have observed the cell surface association of endogenous α_2A_-adrenoceptors in quiescent arteriolar microVSM. However, the expression (mRNA and protein) of these α_2A_-adrenoceptors was inhibited by serum. For example, quiescent microVSM showed increased α_2A_-adrenoceptor biological activity which was significantly reduced in cells stimulated with 10% FBS. In contrast, serum significantly increased α_2C_-adrenoceptor expression (mRNA and protein) and cell-surface-associated biological activity [27]. The current study, therefore, supports the differential regulation of α_2_-adrenoceptors A and C, specifically coupled to Rap1A GTPase signaling, which agrees with previous studies which showed the activation of Rap1 GTPase by serum [45]. The current study further shows that this differential regulation is mediated by the carboxyl-termini of these receptor subtypes.

Similarly, the differential regulation of α_2_-adrenoceptors and function were observed in cutaneous blood vessels during cooling. Specifically, α_2A_-adrenoceptor activity predominated at warm temperatures (37 °C) in mouse cutaneous arteries, whereas the cold-induced (28 °C) vasoconstriction was mediated by α_2C_-adrenoceptors [20]. The transient transfection studies showed similar cell surface localization and functional responses of α_2A_-adrenoceptors at 37 °C and 28 °C and, therefore, supported the distinct role of α_2C_-adrenoceptors in modulating vessel tone [24].

Previous studies have reported α_2A_-adrenoceptor single nucleotide polymorphism which contributes to increased receptor expression affecting insulin secretion and increasing type 2 diabetes risk [46]. Similarly, two recent genome-wide association studies (GWAS) have identified α_2_-adrenoceptor, particularly the α_2A_-adrenoceptor subtype variant rs7090046 (which affects expression in microvascular smooth muscle cells in blood vessels), as a major risk factor in primary Raynaud’s in a subset of patients having the clinical manifestation of Raynaud’s phenomenon, suggesting that the α_2A_-adrenoceptor modulates the susceptibility to Raynaud’s [47,48]. The combined effect of α_2A_-adrenoceptor variant rs7090046 and α_2C_-adrenoceptors may therefore contribute to disease severity and clinical symptoms. 

Additionally, Tervi et al. also identified a variant rs3918226 of nitric oxide 3 (NOS3), which potentially affects gene transcription and is associated with essential hypertension [47,49]. NOS3 is predominantly expressed in endothelial cells which line blood vessels and regulates the production of vasodilator nitric oxide (NO).

NO is known to increase cyclic guanosine monophosphate—cGMP-dependent protein kinase (cGMP-cGK) in VSM and inhibit RhoA [50,51] and, therefore, can potentially inhibit α_2C_-adrenoceptor translocation. Indeed, preliminary studies in our laboratory using cGMP analogs support this possibility (MA Chotani, unpublished observations). Further, NOS substrate L-arginine and phosphodiesterase-5 inhibitors (that increase cGMP) have been reported to improve the dilation of microcirculation and have been suggested treatments for stage-dependent Raynaud’s [52,53,54,55,56,57]. Altogether, these studies allow the formulation of a conceptual model summarized in Figure 11. 

In ongoing parallel studies, we have further explored the findings of the protein–protein docking of the α_2C_-adrenoceptor C-terminus and filamin-2, which identified critical amino acid residues of α_2C_-adrenoceptor interacting with filamin-2 residues. These include three arginine residues at amino acids 454, 456, and 461 which interact with negatively charged residues in filamin-2 (E2004, E2059, and D2060), respectively. Also, lysine K449 in α_2C_-adrenoceptor is stabilized by aspartic acid D2032 in filamin-2 [43]. The virtual screening of small molecule libraries for docking to the hotspot regions of the α_2C_-adrenoceptor-filamin-2 complex, which could potentially disrupt these interactions, was performed. These studies have identified five lead compounds that could potentially interfere with receptor protein interactions. We have successfully tested one lead compound in established primary vascular smooth muscle cultures which express endogenous α_2C_-adrenoceptors, which were examined using immunofluorescence microscopy and functional assays [32]. It is anticipated that the combination of these approaches will provide alternative and new approaches to target intracellular α_2C_-adrenoceptors.

## 4. Materials and Methods

### 4.1. Recombinant DNA Used in the Study

#### 4.1.1. Plasmid Constructs

##### For Domain-Swapping Studies

The α_2A_-adrenoceptors used in previous studies also harbor the FXXXFXXXF motif in the C-terminus similar to the α_2C_-adrenoceptors but not the arginine-rich region. Further, the α_2A_-adrenoceptor C-terminal region did not show a specific interaction with filamin-2 [33]. Also, the α_2C_-adrenoceptor arginine region when mutated to alanines (R454–458→A454–458) abolished the retention signal and diminished the intracellular retention of α_2C_-adrenoceptors. This mutation did not interact with filamin-2 and behaved like the α_2A_-adrenoceptors and was expressed on the cell surface [33]. In this scenario, the ER export motif may mediate cell surface delivery. Domain-swapping experiments were performed to further test this model. Full-length amino termini HA-tagged chimeras of murine α_2_-adrenoceptors (α_2A_- and α_2C_-adrenoceptors) were generated, which are summarized in Figure 12. The murine and human receptors are highly homologous (particularly, the regions of interest are identical) and show similar physiological responses to cooling and to cAMP-EPAC-RAP1A signaling [24,27].

A BamHI restriction enzyme site was introduced in the coding region of α_2A_- or α_2C_-AR in HA-tagged vectors using site-directed mutagenesis with forward and reverse oligonucleotide primers (QuickChange XL site-directed mutagenesis kit, Stratagene, La Jolla, CA). This allowed the release of a DNA fragment harboring the carboxyl terminus regions, which were agarose gel purified and ligated to BamHI linearized expression plasmids for α_2A_- or α_2C_-adrenoceptors, transformed in bacteria, and colonies screened utilizing miniprep DNA. DNA that showed the successful ligation of carboxyl-terminus fragments was further screened by DNA sequencing for the confirmation and selection of the correct construct. Finally, the BamHI restriction site in each construct was removed, and the wild-type DNA sequence was restored using site-directed mutagenesis. 

The constructs (referred to as Chimera 1, HA-α2A-2C and Chimera 2, HA-α2C-2A) generated by this approach were examined for any gross deletions by restriction enzyme digestions and agarose gel analyses and were finally confirmed by DNA sequencing.

##### Hemagglutinin-Tagged Receptors

The DNA constructs, including murine amino-terminal hemagglutinin (HA)-tagged full-length wild-type α_2A_- and α_2C_-adrenoceptors in pcDNA3 expression vector have been described in previous studies [21,24]. These constructs are referred to as HA-α_2A_-AR and HA-α_2C_-AR.

##### Green Fluorescent Protein (GFP)-Tagged α_2C_-Adrenoceptor

Green fluorescent protein-tagged human wild-type α_2C_-adrenoceptor has been described previously [33]. The α_2C_-adrenoceptor coding region was fused in frame with enhanced green fluorescent protein (EGFP), with six amino acids of spacer region between the two genes. This construct is referred to as α_2C_-AR-GFP.

##### Rap1A Expression Construct

Rap1A-63E (constitutively active; Rap1A-CA) plasmid DNA and the backbone plasmid (pcDNA) were used in α_2_-AR studies described previously [33,45].

##### For Peptide Expression and Uptake Studies

These constructs were generated by PCR amplification using primers with restriction enzyme overhangs, utilizing wild-type receptors as a template (Figure 13), and were ligated to compatible restriction enzyme sites in a bacterial expression pTAT-HA plasmid. This plasmid is a T7 RNA polymerase-based expression vector for use with an isopropyl β-D-1-thiogalactopyranoside (IPTG)-inducible expression system and confers ampicillin resistance to bacteria for the selection of transformed cells [58]. The peptides of interest can be expressed using this plasmid and include the in-frame fusion of 6X-Histidine (HHHHHH) for nickel column purification, TAT protein transduction domain (YGRKKRRQRRR) for cellular delivery, and hemagglutinin (HA, YPYDVPDYA) tag for the detection of peptides by anti-HA antibody in Western blotting and immunofluorescence microscopy. The delivery of TAT-conjugated peptides has been previously reported to occur in a cell-type independent manner [59], through micropinocytosis [60].

The correct constructs were identified through restriction enzyme digestions of miniprep DNA, and the sequence was confirmed through DNA sequencing. The constructs were designated as pTAT–HA–α2A–AR^438–450^ and pTAT–HA–α2C–AR^449–462^.

The specific amino acid sequences of the peptides along with the predicted molecular mass are summarized in Figure 14. The ProtParam tool was used to estimate the molecular mass of each peptide (https://web.expasy.org/protparam; accessed on 15 February 2022).

### 4.2. Peptide Expression

For the bacterial expression of peptides, the purified constructs were transformed in bacterial strain BL21(DE3) derivative STAR (ThermoFisher Scientific, Waltham, MA, USA) and conditions were optimized for enhanced peptide expression.

The BL21(DE3)-STAR strain is a genetically modified strain which lacks the ability to degrade mRNA. This strain contains a mutated rne gene (rne131) which is responsible for the production of truncated RNAse E enzyme which allows the mRNAs to be stable for elongated periods of time in the cell, thus facilitating protein expression. 

In order to optimize the expression conditions of the peptides for successful purifications, multiple expression conditions were tested, which included varying IPTG concentrations, incubation temperatures, and the duration of incubation. The optimized conditions included inductions at an IPTG concentration of 0.5–0.7 mM, at 18 °C for 16 h.

After induction, cultures were centrifuged and pellets were frozen for SDS-PAGE and Western blot analysis by transferring to polyvinylidine fluoride (PVDF, 0.2 µm, Bio-Rad, Hercules, CA, USA) membrane for blotting utilizing anti-HA primary antibody (monoclonal 2-2.2.14, ThermoFisher Scientific, Waltham, MA, USA) against the peptide tag and secondary goat anti-mouse HRP (ThermoFisher Scientific, Waltham, MA, USA). The signal was visualized by chemiluminescence (WesternSure Reagent, LI-COR Biosciences, Lincoln, NE, USA) and detected using digital imaging (C-DiGit**^®^** Blot Scanner, LI-COR Biosciences, Lincoln, NE, USA).

#### Peptide Purification with Immobilized Metal Ion Chromatography

Peptides were purified through immobilized metal ion chromatography (IMAC), manually using nickel resins, as well as through a pre-packed column. For manual purification, HisPur Ni-NTA resins were used (Thermo-Fisher Scientific, Waltham, MA, USA), and for column purification, a HisTrap column (Cytiva, Global Life Sciences, Marlborough, MA, USA) was used following the manufacturer’s recommendations. The purified proteins were analyzed on SDS-PAGE followed by Western blotting.

### 4.3. Mammalian Cell Culture 

#### 4.3.1. MicroVSM

Human microVSM were derived from dermal arterioles and have been characterized in our previous studies on α_2_-adrenoceptors [27,28,29,33,45]. Murine microVSM were derived from C57BL/6 wild-type mouse tail artery explants, which have been characterized in our previous studies for α_2C_-adrenoceptor trafficking and function [29,32,33]. These cells were derived from the mid-distal portion of the tail artery which shows α_2C_-adrenoceptor expression and α_2C_-AR-mediated vasoconstriction responses [20,29].

#### 4.3.2. NIH/3T3

NIH/3T3 mouse embryo fibroblast cell line (American Type Culture Collection (ATCC) No. CRL-1658^TM^) was available at the institute (PCMD) and kindly provided by the PCMD Bio-Bank Facility. NIH/3T3 cells were grown in ATCC-recommended medium, Dulbecco’s Modified Eagle Medium DMEM (high glucose/L-glutamine/pyridoxine HCl, 110 mg/L sodium pyruvate, 1.5 g/L sodium bicarbonate, supplemented with donor bovine serum fortified with iron) (Gibco, Grand Island, NY, USA).

### 4.4. Transient Transfections and Co-Transfection Studies

Transfections of NIH/3T3 cells were performed by plating 50,000 cells on glass coverslips which were placed in 24-well culture plates. Cells were transfected 24 h after plating. Cells were transfected using Lipofectamine^TM^ 3000 transfection reagent (Thermo Scientific, Waltham, MA, USA) using Opti-MEM^TM^ reduced serum medium (Thermo Scientific, Waltham, MA, USA), following the manufacturer’s recommendations. Lipofectamine^TM^ 3000 (0.9–1 µL) was used with 1 µg of total DNA, with a 24 h time period of transfections in Opti-MEM^TM^ reduced serum medium, after which the medium was washed once with 0.5% DMEM, and cells were allowed to recover for 48 h in 0.5% DMEM.

#### Transient Co-Transfections and Peptide Studies

After a 24 h period of transfections in Opti-MEM^TM^, the medium was washed once with 0.5% DMEM, and cells were allowed to recover for 24 h in 0.5% DMEM. Subsequently, 25 µL of purified peptides were added to the medium, and cells were incubated for another 24 h. The cells were fixed for 20 min at room temperature in 4% paraformaldehyde and stored at 4 °C. 

### 4.5. Immunostaining and Imaging

Fixed cells were permeabilized (0.1% Triton X–100, 15 min), placed in a blocking solution (ReadyProbes^TM^ 2.5% normal goat serum, Thermo Scientific, Waltham, MA, USA) for 90 min at 37 °C, followed by incubations with the primary antibody (anti-HA monoclonal 2-2.2.14, used at 1:250 dilution, 90 min, at 37 °C), secondary antibody (Alexa Fluor 568 goat anti-mouse, used at 1:200 dilution, 90 min, at 37 °C), and finally, with DAPI nuclear stain (Thermo Scientific, Waltham, MA, USA, 1 µg/mL, 30 min, room temperature). The coverslips were mounted on glass slides with anti-fade mounting media (ProLong^TM^ Antifade Mountant, Thermo Scientific, Waltham, MA, USA). For endogenous α_2C_-adrenoceptors, a 1:80 dilution of the primary antibody (monoclonal anti-α_2C_-adrenoceptor, clone S330A-80, 1 μg/μL stock, Invitrogen, Carlsbad, CA, USA) was used. 

#### Imaging and Receptor Quantification

Cells were visualized using fluorescence microscopy (Nikon Eclipse T*i*2 fluorescence microscope with DS-Ri2 camera, Nikon Corporation, Tokyo, Japan), and the mean fluorescence intensity of the surface HA- or GFP-tagged or endogenous α_2_-adrenoceptors at the cell boundary was quantitated by Nikon NIS Elements software (version 5.10) and the region of interest (ROI) tool at 4–6 random ROI on the cell boundary per cell. The fluorescence was compared between various transfected or non-transfected cells by using the identical power level and fluorescence channel camera settings. 

### 4.6. Assessing the Effect of Peptide C on Receptor Function 

This was achieved by cAMP assays, established in our previous studies [29,32]. Briefly, microVSM (100,000 cells/well) were plated in a 6-well plate in 10% growth medium and incubated at 37 °C for 24 h in a 5% CO_2_ humidified incubator. Subsequently, cells were placed in 0.5% quiescent media for 5 days, after which Peptide C was added to the medium for 4–6 h (for uptake), the medium removed, and cells washed with 0.5% quiescent medium, followed by the addition of solvent (H_2_O, control) or 8-pCPT-2′-O-Me-cAMP (100 µM, 14 h). Cells were pre-treated with the phosphodiesterase inhibitor IBMX (225 µM) for 30 min at 37 °C, followed by treatment with the α_2_-adrenoceptor agonist UK-14,304 (10 nM) for 1 min, then with forskolin (10 µM, 5 min). After one ice-cold PBS wash, cells were lysed in 0.1M HCl and stored at −20 °C. The frozen cell lysates were thawed and centrifuged at 600× *g* for 10 min, and the supernatant was used to assess intracellular cAMP in the presence or absence of Peptide C using a competitive ELISA kit (Enzo Life Sciences, Farmingdale, NY, USA), following the manufacturer’s guidelines.

### 4.7. Statistical Analysis

The significance of relative difference was assessed by statistical analysis using Student’s *t*-test. The graphs were created using GraphPad Prism 5 for Windows (GraphPad, Boston, MA, USA) and the mean values were expressed as the ±SEM. Differences were considered as statistically significant when *p* ≤ 0.05.

## 5. Conclusions

In this study we have tested the role of the carboxyl-terminus of α_2_-adrenoceptor subtype C in receptor trafficking from intracellular store to the cell surface via protein–protein interactions with the actin-binding protein filamin-2. By using domain-swapping and decoy peptide approaches to interfere with protein interactions, we have established the proof of principle to target intracellular α_2C_-adrenoceptors to modulate biological activity.

## Figures and Tables

**Figure 1 ijms-24-17558-f001:**
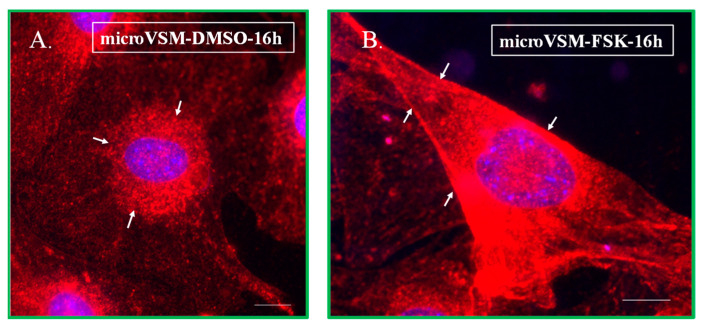
Endogenous α_2C_-adrenoceptor translocation in murine microVSM. The effect of solvent DMSO (16 h, (**A**)) compared with the effect of the adenylyl cyclase activator forskolin (16 h, (**B**)) on endogenous α_2C_-adrenoceptor translocation when added to quiescent primary murine microVSM, determined by immunofluorescence (Alexa Fluor 568 red, α_2C_-adrenoceptors; DAPI blue, nucleus). A similar response is seen in human-derived microVSM [29]. The arrows in (**A**) point to intracellular receptors which are localized in the perinuclear region, whereas the arrows in (**B**) point to receptors on the cell surface (referred to as the cell boundary). Scale bar = 50 µm, 60× oil objective.

**Figure 2 ijms-24-17558-f002:**
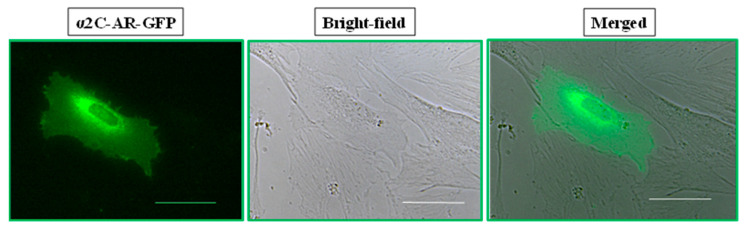
Transient transfection of α_2C_-AR-GFP in murine microVSM. Recombinant α_2C_-adrenoceptor tagged with green fluorescent protein (GFP) transiently transfected in murine tail artery explanted vascular smooth muscle cells. The intracellular localization of the receptor can be seen in the perinuclear region under quiescent, unstimulated conditions, similar to endogenous α_2C_-adrenoceptors in Figure 1A. Scale bar = 50 µm, 40× objective.

**Figure 3 ijms-24-17558-f003:**
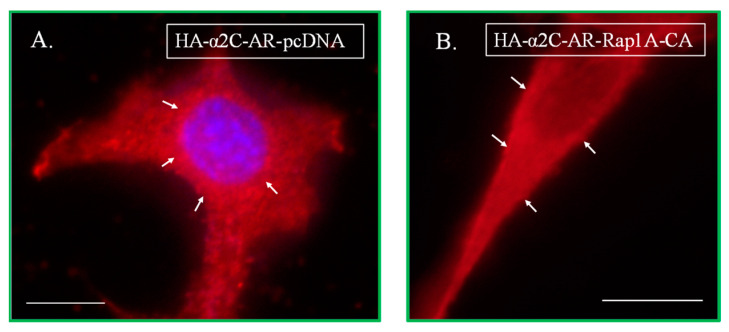
Transient co-transfections in NIH/3T3 cells using HA-α_2C_-adrenoceptors (HA-α_2C_-AR). Cells were co-transfected with HA-α_2C_-AR along with an empty expression vector (pcDNA, (**A**)) or with constitutively active Rap1A (Rap1A-CA, (**B**)). The arrows in (**A**) point to intracellular receptors which are localized in the perinuclear region, whereas the arrows in (**B**) point to receptors on the cell surface (referred to as the cell boundary), similar to the endogenous receptors in Figure 1. Scale bar = 10 µm, 60× oil objective.

**Figure 4 ijms-24-17558-f004:**
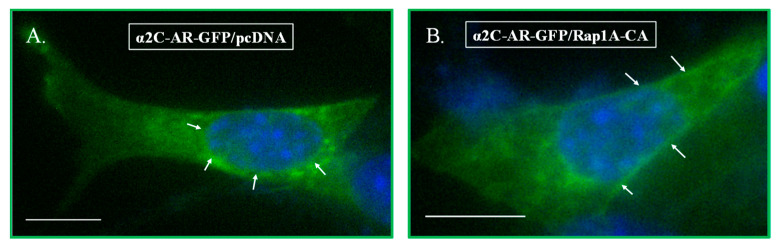
Transient co-transfections in NIH/3T3 cells using α_2C_-AR-GFP. Cells were co-transfected with α_2C_-AR-GFP along with an empty expression vector (pcDNA, (**A**)) or with constitutively active Rap1A (Rap1A-CA, (**B**)). The arrows in (**A)** point to intracellular receptors which are localized in the perinuclear region, whereas the arrows in (**B)** point to receptors on the cell surface (referred to as the cell boundary), similar to the endogenous receptors and recombinant HA-tagged receptors in Figure 1 and Figure 3, respectively. Scale bar = 10 µm, 60× oil objective.

**Figure 5 ijms-24-17558-f005:**
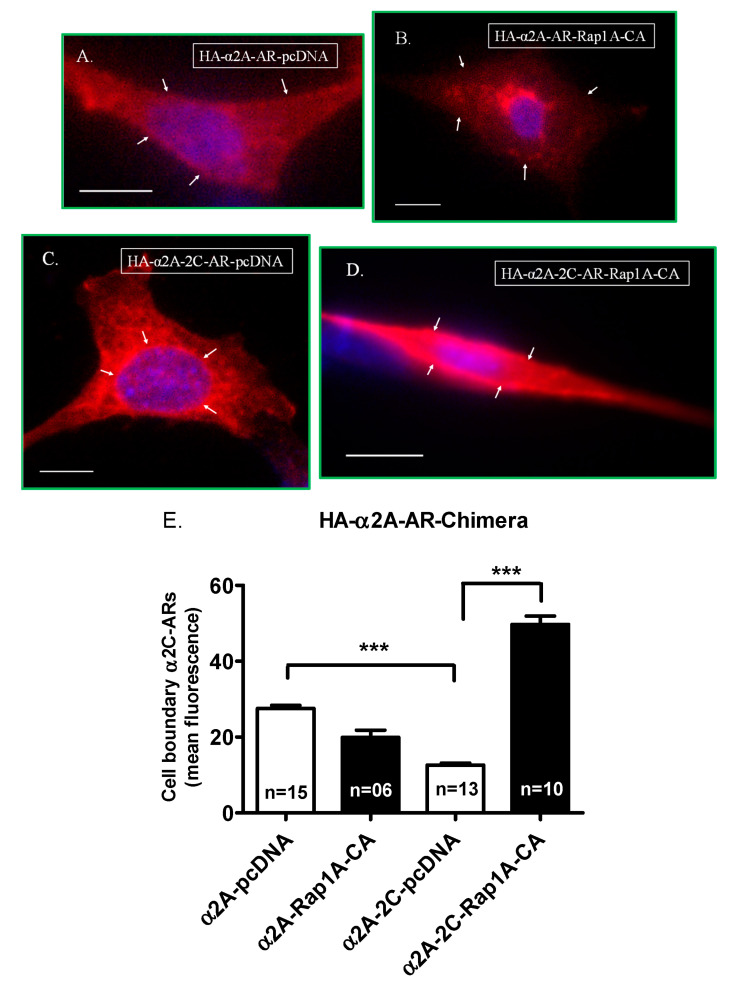
Transient co-transfections of wild-type α_2A_-adrenoceptors and α_2A_-_2C_-AR chimera. Cells were co-transfected with α_2A_-adrenoceptor and the control expression vector pcDNA (**A**) or with constitutively active Rap1A (Rap1A-CA) (**B**). Similarly, the α_2A_-_2C_-AR chimera was co-transfected with pcDNA (**C**) or with Rap1A-CA (**D**). The receptors at the cell boundary were assessed by quantitating the mean fluorescence intensity at six random regions of interest (ROI) on the cell boundary per cell. The data from the (n) number of cells analyzed for each construct are shown; *** *p* < 0.0001 (**E**). The arrows point to receptor localization under the tested conditions. The scale bar = 10 µm, 40× objective. The data presented are available in the Appendix A.

**Figure 6 ijms-24-17558-f006:**
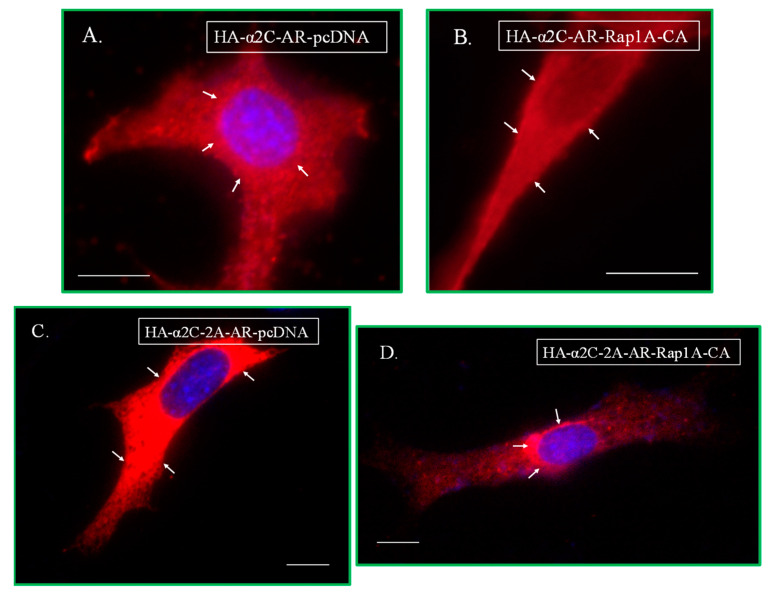
Transient co-transfections of wild-type α_2C_-adrenoceptor and α_2C_-_2A_-AR chimera. Cells were co-transfected with α_2C_-adrenoceptor and the control expression vector pcDNA (**A**), or with constitutively active Rap1A (Rap1A-CA) (**B**). Similarly, the α_2C_-_2A_-AR chimera was co-transfected with pcDNA (**C**) or with Rap1A-CA (**D**). The receptors at the cell boundary were assessed by quantitating the mean fluorescence intensity at six random regions of interest (ROI) on the cell boundary per cell. The data from the (n) number of cells analyzed for each construct are shown; *** *p* < 0.0001 (**E**). The arrows point to receptor localization under the tested conditions. The scale bar = 10 µm, 60× oil objective. The data presented are available in the Appendix A.

**Figure 7 ijms-24-17558-f007:**
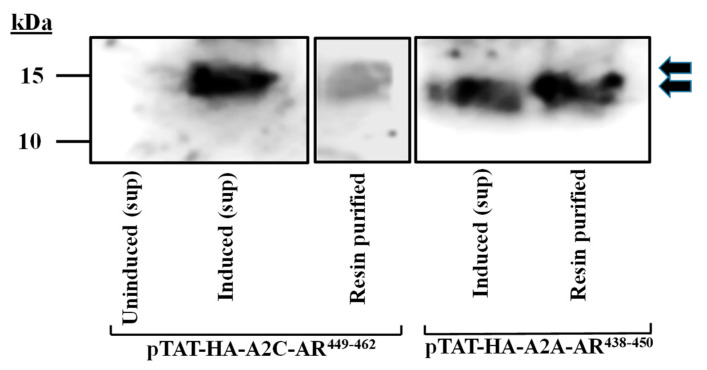
Western blotting and the detection of expressed peptides. SDS-PAGE peptide separation of pTAT–HA–A2A–AR^438–450^ and pTAT–HA–A2C–AR^449–462^ expressed in BL21 (DE3)-STAR on 16% resolving gel. The peptides were detected using an anti-HA monoclonal antibody (clone 2-2.2.14), targeting the HA-tag on the peptides. Supernatant (sup).

**Figure 8 ijms-24-17558-f008:**
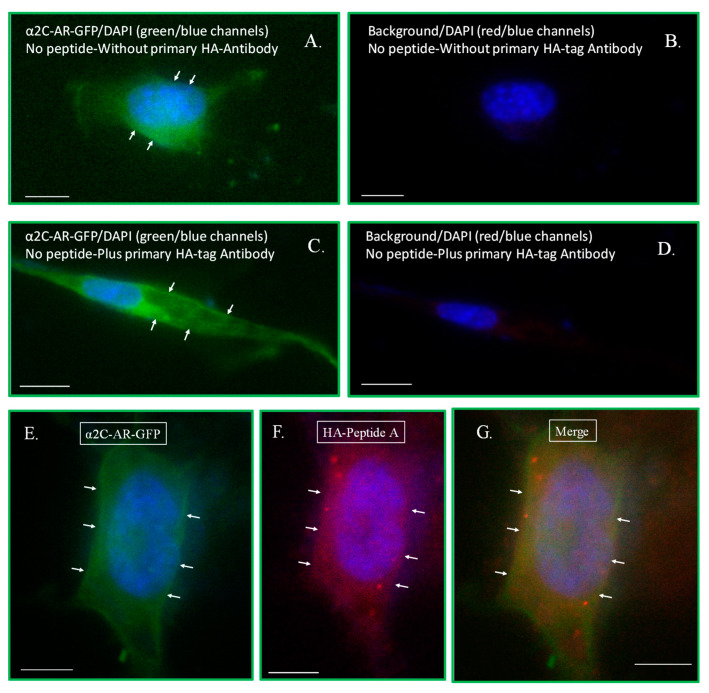
Peptide delivery and localization. The peptide staining was distinct from the background signal, which was assessed for α_2C_-AR-GFP/Rap1A-CA co-transfected cells without peptides and without or with primary anti-HA antibody (**A**–**D**). The cellular uptake of both peptides was observed, with localization in the cytosol and/or the perinuclear region for pTAT–HA–A2A–AR^438–450^ (Peptide A) (**E**–**G**) and pTAT–HA–A2C–AR^449–462^ (Peptide C) (**H**–**J**). The arrows point to α_2C_-AR-GFP localization. The scale bar = 10 µm, 60× oil objective.

**Figure 9 ijms-24-17558-f009:**
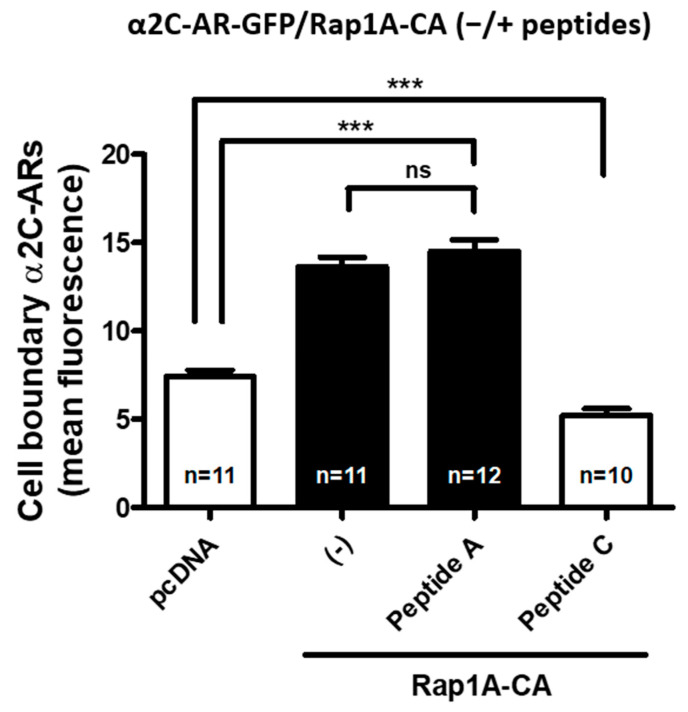
Peptide specificity to inhibit receptor translocation. Transient co-transfection experiments were performed using an α_2C_-AR-GFP reporter along with the control expression vector pcDNA or Rap1A-CA in the absence or presence of pTAT–HA–A2A–AR^438–450^ (Peptide A) or pTAT–HA–A2C–AR^449–462^ (Peptide C). The receptors at the cell boundary were assessed by quantitating the mean fluorescence intensity at six random regions of interest (ROI) on the cell boundary per cell. The data from the (n) number of cells analyzed for each set of experiments are shown; (*** *p <* 0.0001; ns, not significant). The data presented are available in the Appendix A.

**Figure 10 ijms-24-17558-f010:**
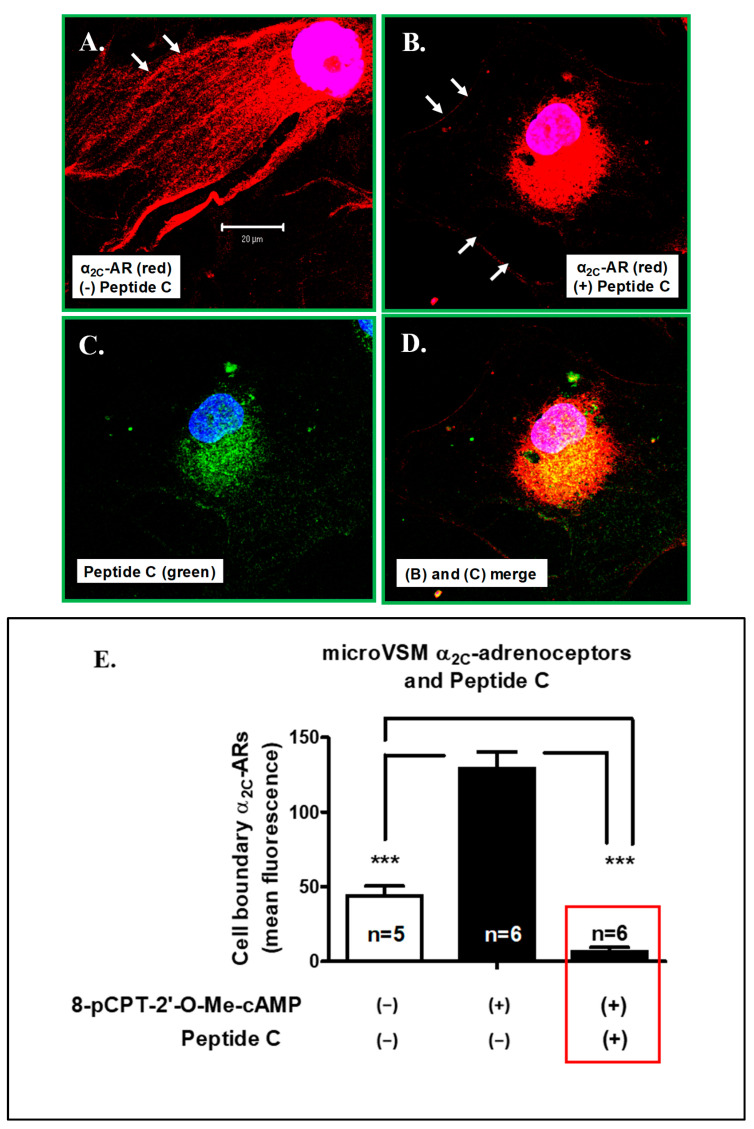
Human microvascular smooth muscle cell Peptide C delivery and effect on endogenous α_2C_-adrenoceptors. (**A**–**D**) Delivery of HA-TAT-α_2C_-AR (Peptide C) to human microVSM and effect on receptor translocation examined by immunofluorescence microscopy (α_2C_-adrenoceptors, Alexa Fluor 568, red; Peptide C, AlexaFluor 488, green; nucleus, blue). Cells were treated with the cAMP analog and Epac-Rap1A activator 8-pCPT-2′-O-Me-cAMP (100 µM, 16 h). The arrows point to cell boundary. Scale bar = 20 µm. The α_2C_-adrenoceptors are perinuclear in the presence of the peptide. (**E**) Quantification of cell boundary α_2C_-adrenoceptors. The receptors at the cell boundary were assessed by quantitating the mean fluorescence intensity at four random regions of interest (ROI) on the cell boundary per cell. The data from the (n) number of cells analyzed for each set of experiments are shown; (*** *p <* 0.0001). In the presence of Peptide C, the cell surface localization of the receptor was below the baseline level in unstimulated cells (red box). (**F**) Assessing microVSM receptor function by measuring intracellular levels of cAMP in 8-pCPT-2′-O-Me-cAMP (100 µM, 14 h) stimulated cells in the absence or presence of Peptide C. The data are corrected for baseline cAMP level and shown as the percent response to forskolin alone, expressed as mean ± SEM for four independent replicates (see Section 4 for details of the assay; * *p* < 0.05). The α_2_-adrenoceptors are Gi-coupled, and activation by the agonist UK, 14,304 inhibits adenylyl cyclase and intracellular cAMP levels. The data presented are available in the Appendix A.

**Figure 11 ijms-24-17558-f011:**
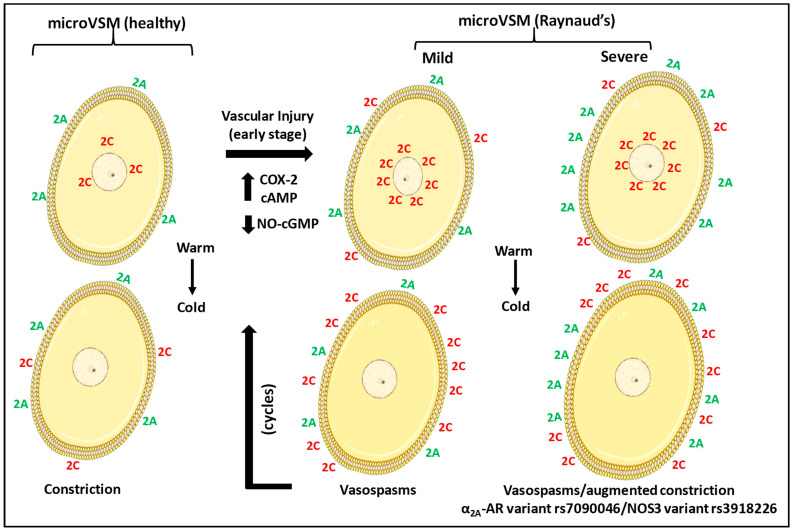
Impact of increased intracellular α_2_-adrenoceptors on VSM constriction at warm and cold temperatures. In the early stage, vascular stress (for example, injury due to cycles of vasospastic attacks or mechanical, followed by inflammation) elevates cyclooxygenase-2 (COX-2), intracellular cyclic AMP (cAMP), transcriptionally increasing α_2C_-adrenoceptors and the intracellular pool of receptors (warm). Cooling mobilizes α_2C_-adrenoceptors to the cell surface, causing vasoconstriction (cold). Vascular injury contributes to the impaired dilator (nitric oxide (NO)—cyclic guanosine monophosphate, cGMP) function of endothelium and increases vasoconstriction via α_2C_-adrenoceptors during the disease process. The cyclic AMP-Rap1A signaling predominates. There is, therefore, a loss of balance between vasodilation and vasoconstriction, tipping in favor of vasoconstriction. Individuals with the variant α_2A_-adrenoceptor (α_2A_-AR) rs7090046 and NOS3 variant rs3918226 have an increased expression of α_2A_-adrenoceptors and reduced vasodilator NO, contributing to a severe clinical condition versus the mild condition in individuals without these variants. The colors denote α_2A_-adrenoceptors (2A, green), and α_2C_-adrenoceptors (2C, red).

**Figure 12 ijms-24-17558-f012:**
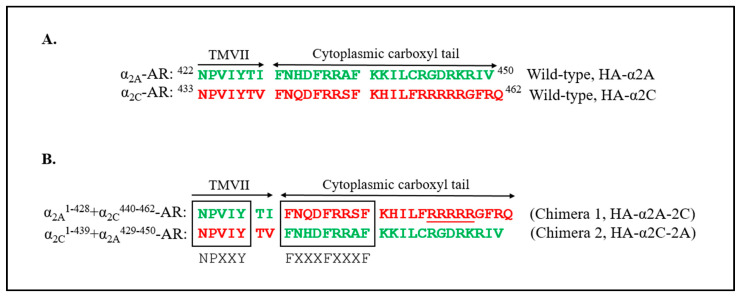
Plasmid constructs for domain-swapping studies. (**A**) Full-length amino terminus HA-tagged wild-type receptor α_2A_- (green) and α_2C_-adrenoceptor (red) and (**B**) chimeras generated for the studies. Carboxyl-termini and part of transmembrane 7 (TMVII) are shown. Putative regulatory regions are indicated, including NPXXY, FXXXFXXXF, and a non-conserved α_2C_-adrenoceptor arginine-rich region (R-454–458). Single letter amino acid codes are shown.

**Figure 13 ijms-24-17558-f013:**
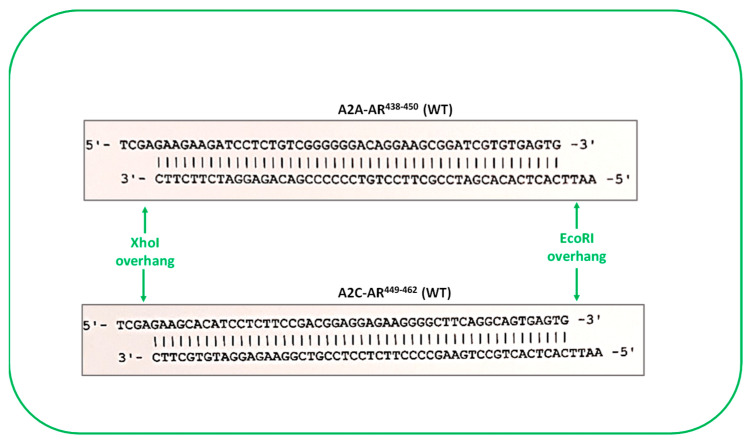
DNA fragments with restriction enzyme overhangs harboring α_2_-adrenoceptor subtypes A and C regions of interest. These fragments were used to generate DNA constructs pTAT–HA–A2A–AR^438–450^ (WT) and pTAT–HA–A2C–AR^449–462^ (WT) for peptide bacterial expression.

**Figure 14 ijms-24-17558-f014:**
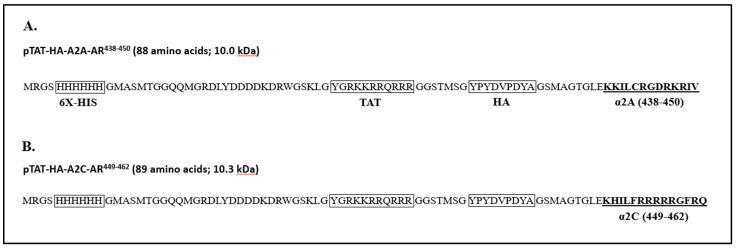
The amino acid sequence of the peptides used in the study. The peptides, including (**A**) pTAT–HA–A2A–AR^438–450^ (WT) and (**B**) pTAT–HA–A2C–AR^449–462^ (WT), were expressed using the bacterial expression vector pTAT-HA. The peptides include the in-frame fusion of 6X-Histidine (HHHHHH) for nickel column purification, TAT protein transduction domain (YGRKKRRQRRR) for cellular delivery, and hemagglutinin (HA, YPYDVPDYA) tag for the detection of peptides by anti-HA antibody in Western blotting and immunofluorescence microscopy.

## Data Availability

The data presented in this study (Figure 5, Figure 6, Figure 9, Figure 10) are available in the Appendix A.

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
