# Peer review of "Inhibiting Intracellular α2C-Adrenoceptor Surface Translocation Using Decoy Peptides: Identification of an Essential Role of the C-Terminus in Receptor Trafficking"

_ijms, 2023, doi:10.3390/ijms242417558_

Round 1
Reviewer 1 Report
Comments and Suggestions for Authors
The manuscript by Raza and colleagues describes the differential localization of the alpha2A and alpha2C receptors on the membrane surface versus perinuclear space when expressed in murine microVSM and NIH/3T2 cell lines. The authors build off of their prior study that found alpha2C receptors traffic to the membrane when exposed to lower temperatures (28C). In this study, they show convincingly, using chimeras, and decoy peptides that the C terminus of the two receptors isoforms endow the receptors with specific localization and opposite translocation to triggered by cAMP-Rap1A GTPase signaling. The results are discussed in the context of Raynaud's phenomenon. The study is novel and written clearly. There are only minor concerns.
1. Data presented in Figure 10 is not formatted corrected and difficult to read. It would appear that Figure 10F is not discussed in the manuscript.
2. It is unclear why the mean fluorescence in Figure 9 is so low when compared to all the other experiments. The comparison between control conditions and with the peptides is difficult to made because because the labeling under basal and stimulation by Rap1A-CA is so low. The authors should explain or discuss this.
typos:
The legends for figure 5 and figure 6 do not include panel E.
line 472 Philipp et al reference should be formatted as a number and included in the references.
Reviewer 2 Report
Comments and Suggestions for Authors
The manuscript describes an already largely accepted methodology to substantiate the hypothesis that the C-terminal of the alpha2C receptor is involved in the translocation of the receptor to the membrane. The experiments are well performed, the results clearly presented, the discussion neatly drawn. They advance a plausible conceptual model to explain the mechanims involved in Raynaud's syndrome although actual proof of the model is not given.
They claim that lead compounds are already tested to alter the translocation (ref.32). They however should be more explicit in their claim. Are these compounds derived from the C-terminal structure of the receptor or are they interfering with other steps in their model. In the first case, they should add the results to line 511-517. In the second case, these paragraph is not relevant to the introduced manuscript.
Minor formal criticism: The authors should make the diagrams in Figure 10 more lisible.
Comments on the Quality of English LanguageThere are some minor errors in the text which could be easily corrected.
